# Does Local Adaptation Impact on the Distribution of Competing *Aedes* Disease Vectors?

Kelly L. Bennett [1,*], William Owen McMillan [1] and Jose R. Loaiza [1,2,3,*]

1 Smithsonian Tropical Research Institute, P.O. Box 0843-03092, Balboa Ancon, Panama; mcmillano@si.edu
2 Instituto de Investigaciones Científicas y Servicios de Alta Tecnología,
P.O. Box 0843-01103, Panama City, Panama
3 Programa Centroamericano de Maestría en Entomología, Universidad de Panamá,
P.O. Box 0843-01103, Panama City, Panama
* Correspondence: bennettK@si.edu (K.L.B.); jloaiza@indicasat.org.pa (J.R.L.)

**Abstract:** *Ae.* (Stegomyia) *aegypti* L. and *Aedes* (Stegomyia) *albopictus* Skuse mosquitoes are major arboviral disease vectors in human populations. Interspecific competition between these species shapes their distribution and hence the incidence of disease. While *Ae. albopictus* is considered a superior competitor for ecological resources and displaces its contender *Ae. aegypti* from most environments, the latter is able to persist with *Ae. albopictus* under particular environmental conditions, suggesting species occurrence cannot be explained by resource competition alone. The environment is an important determinant of species displacement or coexistence, although the factors underpinning its role remain little understood. In addition, it has been found that *Ae. aegypti* can be adapted to the environment across a local scale. Based on data from the Neotropical country of Panama, we present the hypothesis that local adaptation to the environment is critical in determining the persistence of *Ae. aegypti* in the face of its direct competitor *Ae. albopictus*. We show that although *Ae. albopictus* has displaced *Ae. aegypti* in some areas of Panama, both species coexist across many areas, including regions where *Ae. aegypti* appear to be locally adapted to dry climate conditions and less vegetated environments. Based on these findings, we describe a reciprocal transplant experiment to test our hypothesis, with findings expected to provide fundamental insights into the role of environmental variation in shaping the landscape of emerging arboviral disease.

**Keywords:** local adaptation; genomic variation; spatial distributions; biological competition; *Ae. mosquitoes*; climate change

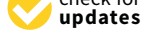

## 1. Introduction

The mosquitoes *Ae. aegypti* and *Ae. albopictus* are the primary vectors of arboviruses to humans, including the most common viruses, dengue (DENV), chikungunya (CHIKV) and Zika (ZIKV), but also Mayaro (MYA) and yellow fever (YF). These diseases are collectively responsible for over 50 million cases and 50,000 deaths every year globally [1,2]. Although there is a vaccine for YF, human cases are on the rise [3] and despite attempts to reduce the burden of disease [4,5], there is currently no effective prevention scheme for other *Aedes*-borne diseases. As a result, the reduction of arboviral disease transmission relies on population control including the larval habitat destruction, adult insecticide spraying and the genetic or biological manipulation of vector populations [6]. The successful implementation of any control method depends on a good understanding of mosquito ecology and evidence-based decisions of when and where this should be implemented [4]. To this end, disease prediction models are an important resource, although their accuracy depends on the specificity of the biological parameters which are input into the model [7]. An important consideration for disease prediction is how *Ae. mosquitoes* may shift their geographical distributions as a result of both biological competition and environmental variation, and how this may influence the disease transmission landscape [4]. Although

shifts in *Aedes* distributions due to climate change have been explored, such models generally disregard the impact of the adaptive potential of local populations on the outcome of inter-specific interactions, which could be a critical driver of biological parameters shaping *Aedes* geographical distributions [8,9]. Furthermore, locally adapted populations can severely impact vector control efforts that are based on a gene drive system, either by promoting or limiting the spread of the target characteristic [10].

## 2. The Displacement of *Ae. aegypti* by *Ae. albopictus* Is Context-Dependent

The global distribution of *Ae. aegypti* has recently been shifting due to the invasion of the ecologically similar Asian tiger mosquito, *Ae. albopictus*. While *Ae. aegypti* has been widespread since its expansion from Africa during the 17th century, *Ae. albopictus* has been expanding from Asia to be globally distributed within the last ~40 years [11]. As a result, *Ae. albopictus* has displaced resident *Ae. aegypti* from many locations including Panama, the South Eastern USA, Madagascar, La Reunion, Mayotte and Bermuda [12–21]. Displacement of *Ae. aegypti* is the expected outcome given that *Ae. albopictus* is a superior competitor under semi-field conditions [14,15]. However, both mosquitoes also often persist together under particular environmental conditions, with this co-existence being stable over time [12,21], which suggests that local adaptation to the environment could play a role in determining displacement or coexistence. Although such studies have found differences in the distribution or abundance of *Aedes* species across geographical regions, few studies have looked at adaptive differences across these populations [17,22,23], and none address the impact of local adaptation on the *Aedes* competitive interaction.

The main mechanisms allowing *Ae. albopictus* to outcompete and displace *Ae. aegypti* are cited as satyrization, where interspecific mating reduces *Ae. aegypti* fitness because hybrids of the latter are not produced, and the biological competition for ecological resources at the larval stage [15,24]. A clear understanding of the factors underpinning the condition dependent displacement of *Ae. aegypti* remain elusive, but environmental conditions appear to play a key role. Previous studies have suggested that the ability of *Ae. aegypti* to persist, despite invasion by *Ae. albopictus*, is associated with dry climate conditions and/or urban environments [9]. These findings are largely based on studies from Florida, which show that *Ae. aegypti* are found in higher abundances in urbanised areas than *Ae. albopictus*, which are more abundant in rural or vegetated settings [9,25]. Differences in species distributions are also associated with changes in rainfall, humidity and temperature across the sampled range [26,27]. Moreover, we [21] recently described the geographical distributions of *Ae. aegypti* and *Ae. albopictus* across the heterogeneous landscape of Panama, including across the Azuero Peninsula, an isolated region subject to a sharp west to east environmental gradient (i.e., >150 km) (Figure 1). In this study, *Ae. aegypti* was shown to persist with *Ae. albopictus* only in regions with a dry and seasonal climate, while displacement was observed in wet tropical areas with a higher humidity and more vegetation. This condition dependent displacement was seen despite both species sharing a similar ecological niche and was hypothesized to result from environmental variation shaping the nature of the species interaction, including the outcome of biological competition.

What permits *Ae. aegypti* to persist with *Ae. albopictus* under some environmental conditions is still unknown. It is thought *Ae. aegypti* could perform better under dry climate conditions because they are better adapted [9]. For example, the eggs of *Ae. aegypti* are more tolerant to higher temperatures and can withstand desiccation for longer periods of time in comparison to the eggs of *Ae. albopictus*, which are able to survive lower temperatures through diapause [9,28]. Innate between-species differences in biological parameters are likely to play a role, but may not provide a full explanation, particularly across tropical environments where temporal variation in environmental variables is limited. In addition to such differences, there is also the possibility that local adaptive differences are seen within populations across a regional scale. Whether mosquitoes are locally adapted is an important consideration when predicting the outcome of species competition since locally adapted species may have a competitive edge and reduce the

successful establishment of invasive competitors [29–32], particularly when confronted by an environmental gradient [33]. Studies using genomic technology provide an invaluable opportunity to link the underlying genomic variation to associated biological parameters. Such an approach provides information on population fitness, providing a comprehensive method to the study of *Aedes* persistence.

### 3. The Genomic Signal of Local Environmental Adaptation in *Ae. aegypti*

There is evidence that populations of *Ae.* mosquitoes are locally adapted to their environments. Previous experimental work has shown that populations of *Ae. aegypti* from divergent climates are locally adapted at the larval stage, because they develop at different rates when reared under the same temperature conditions [23]. In addition, the ability of *Ae. aegypti* to compete with *Ae. albopictus* under laboratory conditions differs among populations [17]. More recently, studies of genomics have revealed that *Aedes* populations are divergent at loci correlated with various environmental parameters [22,34]. This includes across broad spatial scales for *Ae. aegypti*, associated with human population density [35]. Moreover, a recent study by our research group [36] used a landscape genomics approach to reveal that different populations of *Ae. aegypti* across Panama, have differences in genomic variation associated with the environment across a small spatial scale and despite low population structure, i.e., high gene flow. This variation is likely to reflect differential adaption to heterogeneous environments across the country. In particular, we found that genomic loci with a signal of local adaptation differed between wet tropical conditions such as those experienced along the Caribbean coast and the dry tropical conditions typical of the Pacific coast of Panama. Interestingly, the eastern Azuero Peninsula, which is the driest region in Panama (Figure 1), had the least variation in the composition of its putatively adaptive alleles, suggesting these alleles could be fixed in the population. Furthermore, differences in this putatively adaptive genomic variation were largely associated with temperature and vegetation, which are important for the development and survival of *Ae. aegypti* eggs and larvae [37–39]. Of all the loci potentially under natural selection, 128 were consistently identified as outliers and likely to be true positives. These loci provide potential targets for future studies into the local adaptation of *Ae. aegypti*.

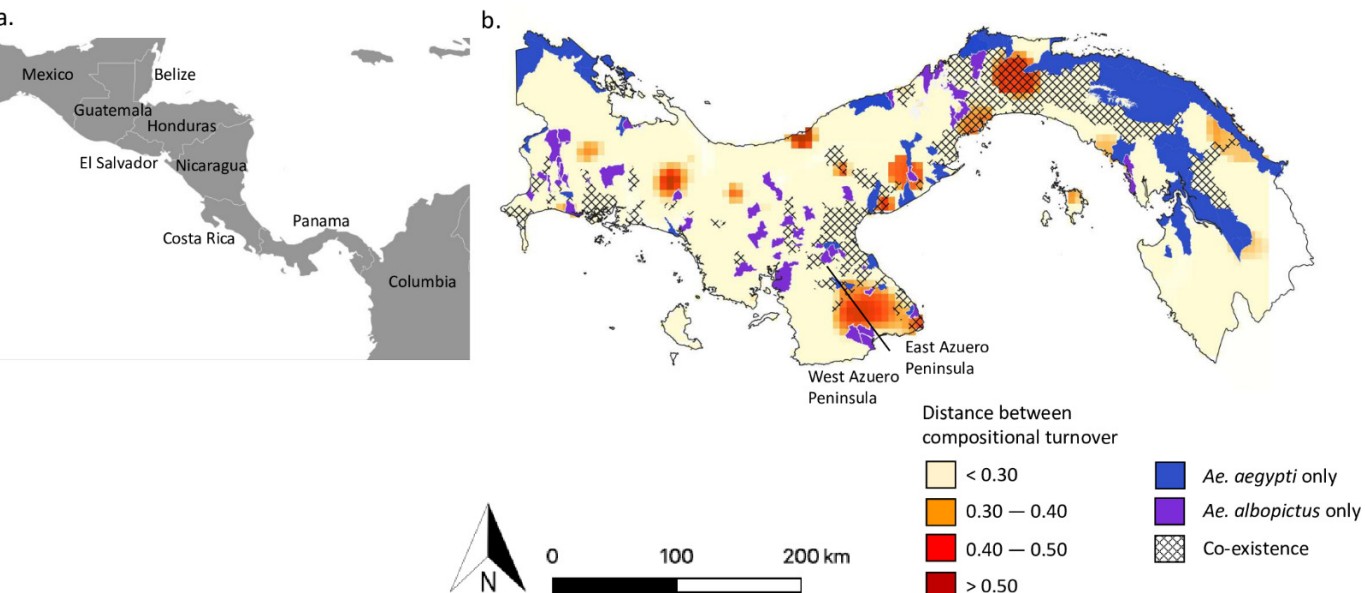

**Figure 1.** Putatively adaptive loci are predicted in *Ae. aegypti* populations within areas of species co-existence. (**a**) The location of Panama within Central America. (**b**) Counties with *Aedes* co-occurrence (dashed areas) are shown based on the recent species distributions as reported by Bennett et al. from 2018 and the Panamanian Ministry of Health (MINSA) in 2017 [21]. The co-occurrence data are overlaid onto a map with red coloured areas representing the potential environmental adaptation of *Ae. aegypti* from Bennett et al. [36]. These areas were identified through a Generalised Dissimilarity Modelling

analysis and represent the difference in allele compositional turnover between a putatively neutral reference loci dataset and a putatively adaptive candidate dataset, using a Procrustes superimposition on their PCA ordinations. A greater predicted distance between the allele composition of the reference and candidate datasets provides a greater indication of populations with adaptive alleles. Purple and blue counties indicate *Aedes* sampling without co-existence.

### 4. Local Adaptation in *Ae. aegypti* Determines Coexistence with *Ae. albopictus*

Given the findings of our two recent studies from Panama that there is condition-dependent displacement, and that some populations of *Ae. aegypti* have putatively adaptive loci in areas of coexistence, we hypothesise that local environmental adaptation of *Ae. aegypti* might allow them to persist in competition with invading *Ae. albopictus*. Since *Ae. aegypti* has been present in Panama for the last ~100 years [40], it is not surprising that they would be locally adapted. However, since *Ae. albopictus* only invaded Panama in 2002 and are still expanding [40], we do not expect their populations to be adapted to the local conditions. Therefore, *Ae. aegypti* could be better able to survive and exploit local habitats than *Ae. albopictus*. If local adaptation plays a role in population persistence, we would expect *Ae. aegypti* to harbour genomic loci with a signal of selection and correlated to the local environmental conditions in regions where both species occur.

We previously identified areas within Panama which are likely to have populations of *Ae. aegypti* with adaptive genomic variation based on Generalised Dissimilarity Modelling analysis. This method compares the turnover in the composition of loci which are not expected to be under selection, i.e., neutral loci, to the compositional turnover in the candidate adaptive loci over geographical space and takes into account differences in the environment [36]. We overlaid our *Aedes* co-occurrence data [21] from Panama onto the graphical representation of adaptive genomic variation of *Ae. aegypti* across the country. Interestingly, this comparison reveals that both *Ae. aegypti* and *Ae. albopictus* tend to coexist in dry tropical regions where *Ae. aegypti* have divergent candidate loci, i.e., where *Ae. aegypti* are potentially adapted to the local environment (Figure 1). This finding is consistent with the prediction that local environmental adaptation contributes to *Ae. aegypti* persistence. For example, *Ae. aegypti* with putatively adaptive loci were found within the dry tropical Pacific regions of Chiriquí (David), Coclé, the eastern Azuero Peninsula and provincial Panamá where both species co-occur [21,41]. In comparison, *Ae. aegypti* were no longer found in many areas where these particular set of candidate adaptive alleles were not detected (See Figure 2 for reference to the regions within Panama). Although there was also genetic evidence for local adaptation in the isolated wet tropical region of Bocas del Toro and Costa Abajo near Colon, it is unknown whether this variation will allow *Ae. aegypti* to resist invasion by *Ae. albopictus*, given that *Ae. albopictus* was only recorded in Costa Abajo in 2018 and has not yet reached Bocas del Toro. Although tantalising, the patterns observed here are correlational and based on genomic information alone without the measurement of biological parameters. The hypothesis that local adaptation to the environment in *Ae. aegypti* influences their persistence on invasion by *Ae. albopictus* requires further confirmation within an experimental framework.

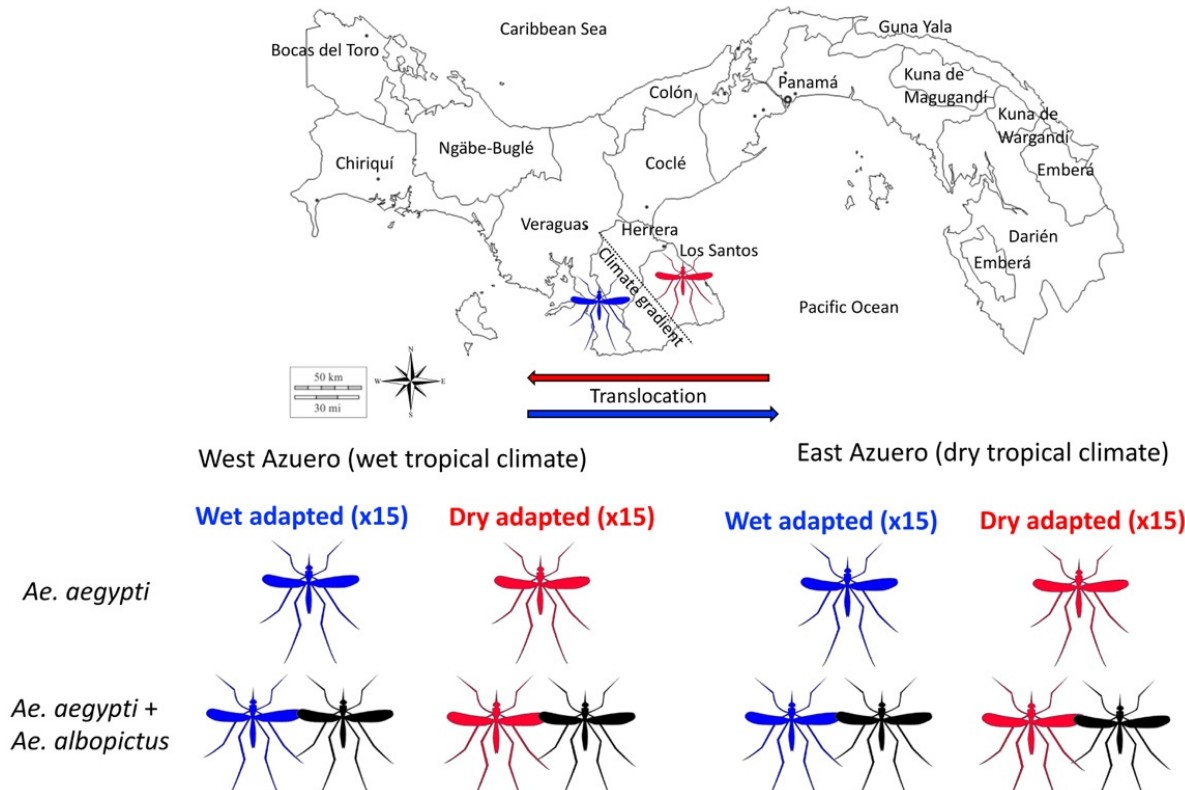

**Figure 2.** Reciprocal transplant experiment to test the hypothesis that local adaptation promotes the persistence of *Ae. aegypti* in the presence of a competitor. Laboratory generated populations of "dry adapted" *Ae. aegypti* sourced from the dry and hot East Azuero Peninsula (red mosquito) could be tested for fitness phenotype and genotypes as they develop both in their local environment and when transplanted across a climate gradient to the wetter region of the Southwest Azuero Peninsula. Similarly, "wet adapted" *Ae. aegypti* from the West Azuero Peninsula (blue mosquito) are transplanted to the dry East Azuero Peninsula as well as being tested under local conditions. In addition, the fitness phenotypes and genomic variation of both wet and dry adapted *Ae. aegypti* should be characterised with and without its competitor, *Ae. albopictus* (black mosquito), under both local and contrasting climate conditions.

## 5. An Experimental Framework to Test for Local Adaptation in the Presence of a Competitor

Given that both *Aedes* species coexist in the dry tropical eastern Azuero Peninsula of Panama, where *Ae. aegypti* appear to be locally adapted, but not in the wet tropical conditions of the southwestern Azuero Peninsula, the hypothesis could be tested with a reciprocal transplant experiment (Figure 2). First, to confirm whether *Ae. aegypti* are truly adapted to the local environmental conditions, the impact of different climate conditions on a range of fitness phenotypes should be determined. The expectation is that *Ae. aegypti* adapted to the local environmental conditions will have a higher fitness within their source environment and experience a lower fitness on translocation to divergent environmental conditions. The experimental design requires at least F1 *Ae. aegypti* acquired from the controlled mating of both putatively 'dry adapted' (i.e., East Azuero) and 'wet adapted' (i.e., Southwest Azuero) laboratory colonies sourced from wild populations. This is both to ensure parentage and so that all samples are subjected to the same ambient conditions before their placement within the experiment, i.e., to ensure that the observed response to the environment results from genetic heritage and not environmental plasticity. Eggs obtained from the matings will form the basis of the experiment, whereby replicate batches of mosquitoes are allowed to develop under semi-field conditions in both the local environment and alternative environment. Differences in observed fitness phenotypes such as egg hatch rate, development time, larval survival and female wing size as a proxy for

fecundity can be determined through comparative statistical analysis. Here, since both populations are tested within their source environment, any natural differences between populations will be apparent. However, it is important to note that care must be taken when extrapolating to other regions, especially where the invasion history of *Ae. aegypti* is unknown. In this circumstance, innate differences in the biological traits of populations, i.e., desiccation resistance, should be tested and/or controlled for.

Second, to test whether the local adaptation of *Ae. aegypti* impacts the outcome of biological competition, these same phenotypic traits should be measured with and without the addition of *Ae. albopictus* within the experimental set up. If the fitness of *Ae. aegypti* is uniformly reduced in the presence of *Ae. albopictus* in both wet and dry environments, then species replacement of *Ae. aegypti* is expected to occur across Panama regardless of climate conditions. However, if *Ae. aegypti* is able to maintain high fitness in the presence of *Ae. albopictus* in dry but not in wet environments, this mechanism could explain the condition-dependent nature of species co-existence. Third, genomic technology targeting the genomic loci we previously identified as putatively under selection should be used to link phenotype to genotype. For example, laboratory experiments using mosquitoes have been used to link quantitative trait loci to phenotypes relating to diapause, viral infection or resistance to insecticides and filarial parasites [42–45]. Informative quantitative analyses [46,47] can be performed to identify which genomic regions are truly impacted by local adaptation and later targeted for functional studies.

## 6. Implications

An understanding of both the function and distribution of adaptive genes will broaden our knowledge of the biological requirements of *Aedes* species important for disease prediction. Improving the accuracy of biological parameters will aid models predicting the outcome of population control, i.e., through the release of *Wolbachia* bacterium, which relies on an increased fitness to the mosquito to promote its spread [10]. Importantly, an understanding of how fitness and adaptive traits are distributed in relation to the environment allows us to consider the adaptive potential when modelling future population responses [48]. With a rise in gene editing strategies and CRISPR technology [49], adaptive genes could be potential targets for population control, either as targets for modification or as genes conferring increased fitness and/or traits that aid the gene drive of disease resistance. Determining whether local adaptation impacts on *Aedes* competitive interactions is important because species co-occurrence could facilitate the emergence of sylvatic arboviral disease. *Ae. albopictus* is an opportunistic feeder, able to utilise a wide range of peri-domestic habitats outside of its native range [50,51] and the species could act as an efficient bridge vector for emergent zoonotic diseases from the forest [51]. The addition of the specialised human commensal *Ae. aegypti* provides the opportunity for any emergent epidemic to spread and be maintained within the urban population [52–55]. Untangling the factors enabling cooccurrence of species will allow us to identify locations with a higher risk of emergent sylvatic disease and act towards preventing new urban outbreaks. In areas where *Ae. albopictus* has displaced *Ae. aegypti*, control strategies currently aimed at targeting *Ae. aegypti* will require modification to account for their wider use of peridomestic habitats and wider range of animal hosts. In addition, gene drive systems which are effective in reducing disease transmission by *Ae. aegypti* [56] will require development for the targeted control of *Ae. albopictus*.

**Author Contributions:** K.L.B., W.O.M. and J.R.L. designed the study. K.L.B. and J.R.L. performed the data analysis and figure preparation. K.L.B. and J.R.L. wrote the manuscript with contributions from W.O.M. All authors have read and agreed to the published version of the manuscript.

**Funding:** This work was sponsored in part by the Government of Panama through MINSA's grant Zika Project (Objective #2) to J.R.L. Support for KLB comes from the Smithsonian's George E. Burch Fellowship and Smithsonian Fellowship Program, the Edward and Jeanne Kashian Family Foundation and Mr Nicholas Logothetis of Chartwell Consulting Group Inc. The National System of Investigation (SNI) of SENACYT currently supports research activities by J.R.L. (157-2017; 16-2020).

**Institutional Review Board Statement:** Not applicable.

**Informed Consent Statement:** Not applicable.

**Data Availability Statement:** All data are previously available in the provided references of Bennett et al. (2021) and Bennett et al. (2021).

**Acknowledgments:** We are thankful to the Panama's Ministry of Environment (Mi Ambiente) and MINSA, the Smithsonian Tropical Research Institute (STRI) and the Instituto de Investigaciones Científicas y Servicios de Alta Tecnología (INDICASAT) for supporting our research and project logistics.

**Conflicts of Interest:** The authors received funding from The Edward M. and Jeanne C. Kashian Family Foundation Inc., and Nicholas Logothetis of Chartwell Consulting. There are no patents, products in development or marketed products associated with this research to declare.

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
