# Peer review of "Does Local Adaptation Impact on the Distribution of Competing Aedes Disease Vectors?"

_climate, doi:10.3390/cli9020036_

Round 1

Reviewer 1 Report

Dear authors,

I have made some comments and suggestions in your manuscript, which I hope you will find useful and constructive.

Author Response

Reviewer one

I have made some comments and suggestions in your manuscript, which I hope you will find useful and constructive

Thank you for your comments which have improved the manuscript.

Responses to comments in manuscript

Do you mean in some environments that Aedes albopictus cannot persist? It's not very clear.

R1: We have clarified the sentence to confirm that we meant Ae. aegypti can persist with Ae. albopictus, despite the latter being a superior competitor.

in Panama?

R2: We have added Panama.

Since its the first occurrence of the Latin name in the main text, please consider writing it in full.

R3: We have amended the Latin name.

attempt to do what?

R4: We have added “to reduce the burden of disease” at this location.

has?

R5: We have corrected the grammar.

Maybe delete "within"?

R6: Deleted.

  1. The genomic....

delete the comma

R8: It has been deleted.

Please provide a reference for this statement.

Ditto

R9: I have added the following reference to address the two comments above.

Eskildsen, G.A.; Rovira, J.R.; Dutari, L.C.; Smith, O.; Miller, M.J.; Bennett, K.L.; McMillan, W.O.; Loaiza, J.R. Maternal invasion history of Aedes aegypti and Aedes albopictus into the Isthmus of Panama: Implications for the control of emergent viral disease agents. PLoS One 2018, 13, e0194874.

Please consider using a more divergent color palette, as it is very difficult to distinguish between the different green gradients. Please also put Panama (I guess it's Panama depicted here) in geographical context. Also please make the map color-blind friendly.

R10: I have changed the map to a red/blue colour scheme to be colour blind friendly. I have placed a map of the Americas in the figure to place Panama in geographical context.

Why not 0.3 - 0.4, 0.4 - 0.5, > 0.5?

R11: Yes, this would be an improvement. I have changed the legend.

This in Spanish. Please translate it to English.

R12: We have changed this to ‘counties’.

Generalized Dissimilarity Modeling analysis

R13: Corrected.

So since you used GDMs, you should have been able to get a raster prediction map for the entire Panama (if that was your study area). By choosing white coloring for BOTH the unsampled regions and the cells with less than 0.3 compositional turnover, it's not clear for which cells you have data and for which cells you do not have data. Please correct this.

R14: The Reviewer is correct, I have a raster prediction map for the extent of Panama and I have changed the map to show colouring for values below 0.3. I have also removed the box for unsampled areas from the legend to prevent confusion. All unsampled areas are the areas which are not Ae. aegypti only (blue), Ae. albopictus only (purple) or areas of coexistence (dashed area).

This is in contradiction with your legend (0.3).

R15: This was a mistake and has been corrected.

There are no grey cells in the map.

R16: The legend has been amended to reflect the updated map.

Does this mean "corrections"? If yes, corrections to what?

R17: We have corrected this to ‘counties’.

Please show this region in Figure 1.

R18: I have added the East and West Azuero Peninsula to the map in Figure 1.

Please reduce font size in lines 230-237.

R19: This has been checked.

Reviewer 2 Report

The authors review in this manuscript the ecological differences in the distributions of species Aedes aegypti and Aedes albopictus in Panama. They present the hypothesis that local adaptation to the environment is critical in determining the persistence of Ae. aegypti in comparison to the competitor Ae. albopictus. They present a map showing the areas where both species occur. They propose a experiment to test the role of the environment in defining the distribution of both species and particularly, of those areas in common. However, the details provided about the experiment are few. For example, the abstract presents the main hypotheses and results but it does not explain how the analyses will be performed. Further, the abstract it is not clear in that the authors are proposing a research study, not presenting the results from that research. Indeed, the authors do not explain in detail in the manuscript how they plan to perform the reciprocal transplant experiment to test the hypothesis that local adaptation promotes the persistence of Ae. aegypti in the presence of a competitor. Without this information, the paper loses a lot of interest.

Author Response

Reviewer two

The authors review in this manuscript the ecological differences in the distributions of species Aedes aegypti and Aedes albopictus in Panama. They present the hypothesis that local adaptation to the environment is critical in determining the persistence of Ae. aegypti in comparison to the competitor Ae. albopictus. They present a map showing the areas where both species occur. They propose a experiment to test the role of the environment in defining the distribution of both species and particularly, of those areas in common. However, the details provided about the experiment are few. For example, the abstract presents the main hypotheses and results but it does not explain how the analyses will be performed. Further, the abstract it is not clear in that the authors are proposing a research study, not presenting the results from that research. Indeed, the authors do not explain in detail in the manuscript how they plan to perform the reciprocal transplant experiment to test the hypothesis that local adaptation promotes the persistence of Ae. aegypti in the presence of a competitor. Without this information, the paper loses a lot of interest.

R20: The Reviewer is correct that not enough detail was present in the text. We have modified the abstract to make it clear that we are presenting a hypothesis/research study and included information on the type of experiment we propose. We have added significantly more detail on the experiment we intend to perform in section 5 and hope this makes the study we are proposing much clearer to the reader. See this addition in Page 11, lines 268 to Page 12, 301 of the revised manuscript (tracked changes).